# Examining the Impact of Race on Motivational Interviewing Implementation and Outcomes with HIV+ Heavy Drinking Men Who Have Sex with Men

**DOI:** 10.3390/ijerph19073930

**Published:** 2022-03-25

**Authors:** Anthony Surace, David G. Zelaya, Arryn A. Guy, Nadine R. Mastroleo, Ayla Durst, David W. Pantalone, Peter M. Monti, Kenneth H. Mayer, Christopher W. Kahler

**Affiliations:** 1Center of Alcohol and Addiction Studies, School of Public Health, Brown University, Providence, RI 02903, USA; david_zelaya@brown.edu (D.G.Z.); arryn_guy@brown.edu (A.A.G.); ayla_durst@brown.edu (A.D.); peter_monti@brown.edu (P.M.M.); christopher_kahler@brown.edu (C.W.K.); 2Department of Global Health and Population, Harvard Medical School, Boston, MA 02115, USA; kenneth_mayer@brown.edu; 3Department of Psychiatry and Human Behavior, Alpert Medical School, Brown University, Providence, RI 02903, USA; 4Department of Psychology, Binghamton University, Binghamton, NY 13902, USA; nmastrol@binghamton.edu; 5The Fenway Institute, Fenway Health Boston, Boston, MA 02215, USA; david.pantalone@umb.edu; 6Department of Psychology, University of Massachusetts, Boston, MA 02125, USA; 7Department of Infectious Diseases, Beth Israel Deaconess Medical Center, Boston, MA 02215, USA

**Keywords:** motivational interviewing for alcohol use, men who have sex with men with HIV, racial discrimination, mixed methods, qualitative analysis

## Abstract

Motivational interviewing (MI)-based interventions focus on changing behavior through building client motivation. It is unknown how racial mismatch between clients and providers may impact MI implementation and subsequent behavior. We used a mixed methods approach to examine differences in Motivational Interviewing Skill Code (MISC) coded sessions and post-session outcomes between a sample of HIV-positive cisgender men who have sex with men (MSM) participants of an MI-based intervention to reduce heavy drinking who identified as persons of color (POC; *n* = 19) and a matched sample of White participants (*n* = 19). We used quantitative methods to analyze how providers implemented the intervention (i.e., MISC codes) and post-session drinking. We used qualitative analyses of session transcripts to examine content not captured by MISC coding. Quantitative analyses showed that providers asked fewer open-ended questions and had a lower ratio of complex reflections to simple reflections when working with POC participants, but no significant differences were observed in drinking post-intervention between participants. Qualitative analyses revealed that participants discussed how racial and sexual orientation discrimination impacted their drinking. Allowing clients to share their experiences and to explore individually meaningful reasons for behavioral change may be more important than strict adherence to MI techniques.

## 1. Introduction

Motivational interviewing (MI) [1,2,3] is a widely used, highly effective, person-centered form of behavioral intervention designed to elicit and strengthen individuals’ motivations for change [4]. One of the core tenants of MI is facilitating behavior change through enhancing client self-efficacy. MI prioritizes clients’ individuality and autonomy by evoking and strengthening clients’ own verbalized motivations for change and meeting clients’ ambivalence with empathy. Using this approach, providers work with clients to develop specific, actionable goals that are personally meaningful to the client.

In practice, MI sessions entail providers asking open-ended questions (e.g., questions beginning with how, why, or what) to facilitate client discussion. In addition, MI providers employ reflective listening whereby the provider paraphrases the client’s statements to ensure comprehension and further exploration. Reflections can be utilized by the provider to emphasize different components of the client’s statements. Such “complex reflections” are meant to draw the client’s attention to ambivalence around changing their behavior and discrepancies between current behavior and desired outcomes. According to the technical hypothesis of MI [5], these technical proficiencies of MI are hypothesized to be the active ingredients which facilitate client engagement and change talk, which subsequently relate to behavior change [6,7,8,9,10].

Practicing MI, however, involves more than asking questions of and actively listening to clients. In addition to the technical components of MI, there are the relational components. These relational components refer to the provider demonstrating accurate empathy/understanding of their client and respecting their autonomy as an individual [5]. According to the relational hypothesis of MI, these relational components are predicted to foster behavioral change in and of themselves. The rationale is that, by creating a comfortable atmosphere of open dialogue, the provider and client can collaboratively strategize behavioral change [11,12,13].

In order to measure both technical and relational aspects of MI implementation, researchers have established formal coding systems, such as the Motivational Interviewing Skill Code (MISC) [14]. With the MISC, therapy sessions are transcribed. Then, client and provider utterances are parsed by trained coders multiple times. For example, a provider utterance may be coded as an open-ended question (e.g., “How does that make you feel?”). Providers are considered more technically proficient by asking more open-ended questions (relative to close-ended questions) and by providing more complex reflections (relative to simple reflections). The MISC also rates providers on general adherence to the principles of MI. These “global” scores are more akin to relational components and are derived from the coder’s overall impression of the provider’s interactions with the client. Global scores are given via a five-point Likert type scale, with 1 indicating low relational competency (e.g., communicating non-acceptance through judgmental statements) and 5 indicating high relational competency (e.g., communicating acceptance through supportive statements) [12].

MI has been adapted for short-term therapy modalities (1–2 sessions). Such brief motivational interventions (brief MI) have been shown to be highly effective at changing health behaviors [15]. For example MI has been used to reduce cannabis use [16], drinking [17] and increase HIV medication adherence [18]. Further, brief MIs have been adapted for minoritized (i.e., a group that is devalued in society and given less access to its resources) [19] individuals, including cisgender men who have sex with men (MSM) [20,21,22] and Latino/a populations [23,24,25]. However, it is unclear if the way brief MIs are implemented, and by whom they are implemented, within clinical settings may impact the technical or relational components of MI, and thus impact the effectiveness of the intervention. For example, myriad research studies demonstrate that White clinicians’ racial biases influence their diagnoses of their patients/clients [26,27,28,29]. These studies suggest that racial bias can influence not only the ways in which clinicians interact with patients and interpret patients’ symptoms [30]. Therefore, it seems plausible that racially biases held by providers with different identities to their clients may impact how these providers implement brief MI interventions (e.g., by not fully reflecting on clients’ experiences). Such potentially unconscious differences in MI implementation can be especially problematic as it may perpetuate existing mental health/substance use treatment disparities [31,32,33,34]. If such implementation differences do exist, it is critical to identify if such discrepancies prevent clients who identify as people of color (POC) clients from receiving the maximal potential benefit from counseling.

Research suggests that client perceptions of their providers may impact the outcomes of behavioral therapy. For example, incongruity between the identities of providers and clients (e.g., sexual/racial identity) may reduce treatment effectiveness and retention in care [35,36]. This may be particularly troublesome for brief MIs as their short duration does not allow for the development of a deep client-practitioner relationship, which could mitigate the impact of an identity mismatch. As such, it is important to understand how identity match, particularly those identities that may be more saliently perceived by the client and provider (e.g., race), may influence the implementation of brief MIs. Additionally, POC are highly underrepresented in mental health professions [37], creating a disparity in the availability of clinicians with potentially matching racial identities to POC identifying clients. Such underrepresentation may partially account for POC mental health care disparities.

To this end, we conducted a secondary analysis of an already completed randomized clinical trial which utilized a brief MI to reduce alcohol use among a sample of heavy drinking MSM with HIV. We used a mixed methods approach to examine differences in brief MI implementation between White providers and racially matched or mismatched clients in the context of a brief MI for heavy drinking. Using quantitative analyses, we compared provider–client interactions within intervention sessions through transcript review and associated post-intervention alcohol consumption (i.e., drinking quantity) between White providers and POC (racial mismatching) versus White (racial matching) participants. Specifically, we quantitatively compared session transcripts coded using the MISC version 2.5 [14]. We hypothesized that, when client and providers were racially matched, providers would adhere more closely to technical and relational components of MI. Specifically, providers would ask more open-ended questions, deliver a higher complex to simple reflection ratio, and have higher MI adherence ratings relative to sessions in which the provider-client races were different. We also hypothesized that sessions between race-matched individuals would result in more behavior change post-session. Specifically, we expected that participants whose races matched those of their providers (i.e., White) would report fewer past month standard drinks at follow-up relative to race-matched vs. race-mismatched participants. We then employed qualitative analyses to contextualize any potential observed differences in session content between POC and White participants. Specifically, qualitative analyses were used to examine the discussions of stigma/discrimination in the context of drinking not captured by the formal MISC coding.

## 2. Materials and Methods

### 2.1. Participants and Procedures

Data for the current analyses were collected as part of the Reducing Alcohol-related Comorbidities in HIV treatment (REACH) study [21]. The REACH study tested the efficacy of a brief MI to reduce alcohol use among heavy drinking MSM with HIV. All participants (*n* = 180) were recruited from Fenway Health, an urban community health center in Boston, MA which specializes in LGBTQ+ healthcare. Eligible participants were (1) 18+ years old, (2) self-reported recent heavy drinking (i.e., ≥14 drinks per week or drinking ≥5 drinks on a least one occasion in the past 30 days, (3) diagnosed with HIV, (4) self-identified as cisgender gay/bisexual or reported engaging in sex (oral or anal) with a male partner in the past 12 months. All study procedures were approved by the Institutional Review Boards of Brown University and Fenway Health.

### 2.2. Measures

#### 2.2.1. Demographic Information

At an initial study visit, all participants completed a self-administered survey which collected demographic information (e.g., self-identified race).

#### 2.2.2. The Multiple Discrimination Scale

In addition, this survey included the Multiple Discrimination Scale (MDS) [38]. The MDS is a list of 30 discrimination experiences with 10 items each for discrimination based on three subscales: race (α = 0.90), sexual orientation (α = 0.82), or HIV serostatus (α = 0.76). A sample question from the racial discrimination subscale includes “Were you ignored, excluded, or avoided by people close to you because of your race or ethnic background?” A sample question from the HIV discrimination subscale is “Were you rejected by a potential sexual or romantic partner because someone knew or suspected that you are HIV positive?” A sample question from the sexual orientation discrimination subscale is “Were you denied a job or did you lose a job because someone thought you were gay?” For each item, participants endorsed if they had experienced the event within the past year. Each subscale had a possible score range from 0 (no events endorsed) to 10 (all events endorsed). We computed the sum of each subscale, with higher scores representing more past year discrimination events.

#### 2.2.3. Timeline Followback Interview

Following the baseline surveys, current alcohol and drug use were measured via the Timeline Followback interview (TLFB) [39] administered by a research assistant. The TLFB provided the primary drinking outcome measures: number of standard drinks consumed per week and the number of binge drinking episodes (i.e., 5+ standard drinks in a single sitting) in the past month.

### 2.3. Study Procedures

Upon completion of the baseline assessment, participants were randomly assigned to brief MI or to an assessment-only HIV treatment-as-usual (TAU) control. Participants randomized to MI were immediately seen by a staff provider for their brief MI session. Participants assigned to TAU met briefly with a trained provider who explained that their participation for the day was complete. For a detailed description of the intervention, see Kahler et al. [21]. All sessions from this study were transcribed and parsed for provider and client utterances. Transcripts were then coded by a team of trained coders, including the fourth author, using the MISC [14]. Coders received ~60 h of training, including an initial training session and individual and group practice sessions with corrective feedback. For a detailed description of the MISC training, see Kahler et al. [40].

### 2.4. Behavioral Intervention

The brief MI sessions were designed to explore individual participants’ reasons for drinking. During these sessions, providers were able to reflect participants’ ambivalence around their alcohol use and explore reasons for change. Providers also explored how participants’ HIV-positive status impacted their drinking experiences. In addition, providers gave personalized feedback on how participants’ self-reported alcohol use compared to U.S.-based population averages among MSM [21]. Participants also received individualized information on areas of their health that may be impacted by their alcohol use (e.g., liver health and cognitive functioning). Participants who expressed interest in changing their drinking behavior worked collaboratively with providers to identify and implement goals for behavioral change.

Study sessions were highly standardized. Detailed treatment manuals were used to ensure uniform treatment delivery. Providers received ~20 h of training in the study protocol and MI, including readings and role-playing exercises. All intervention sessions were audio recorded, and providers received clinical supervision with a doctoral-level trainer and supervisor to discuss current cases and receive feedback on sessions weekly. Counseling sessions were provided by three master’s-level treatment project staff (two cisgender White men and one cisgender White woman), all of whom had previous experience working in clinical or research contexts with MSM populations and people living with HIV.

### 2.5. Analysis Plan

We used a concurrent mixed methods design to examine brief MI implementation and post-session outcomes (quantitative measures) and within-session content (qualitative analysis using thematic analysis techniques) [41]. Quantitative measures provided information on differences between POC and White participants: (1) baseline reported experiences of identity-based discrimination (race, sexual orientation, HIV status), (2) provider adherence to MI-consistent principles, (3) comparison of providers’ session utterances, and (4) outcomes in drinking behavior across 3-, 6-, and 12-month follow-up. Qualitative thematic analysis entailed exploring session transcripts for themes not covered by MISC coding, which could help contextualize our findings, e.g., identifying/analyzing in-session discussions of racial identity.

#### 2.5.1. Quantitative Analyses

First, we compared POC and White participants for differences in baseline measures (utilizing *t* or *χ*^2^ statistics). For example, we compared POC and White participants’ baseline reported experiences of identity-based discrimination over the prior year. Following baseline comparisons, we compared initial session MISC codes between POC and White participants. Specifically, indices of providers’ relational proficiency (i.e., Global Scores: support of autonomy, empathy, directiveness) when working with POC or White participants. In addition, providers’ technical proficiency was compared between POC and White participants (e.g., ratio open questions and complex reflections). We also compared the coded provider utterances of sessions between POC and White participants. Finally, we used multilevel models to compare participants’ changes in drinking behavior at study follow-up periods (3, 6, and 12 months post-intervention). We used SPSS (version 27) for all quantitative analyses.

#### 2.5.2. Qualitative Analyses

Following our quantitative analyses, we conducted qualitative analyses of the first brief MI session. All sessions used for these analyses were qualitatively coded using thematic analysis (TA) [42,43]. Session transcripts were coded by the first two authors independently. Five randomly selected sessions were double-coded and compared to develop themes, which were then reviewed and finalized by the authors. These sessions were then double coded again using the resultant codebook until reliability was reached. Discrepancies in coding were discussed until 100% consensus was reached. Next, the authors coded the remaining session transcripts independently. All qualitative analyses were conducted via NVivo.

## 3. Results

### 3.1. Description of the Sample

From the total study sample (*n* = 180), roughly 25% (*n* = 45) self-identified as a POC (i.e., Black/African American, American Indian/Alaskan Native, Asian and/or non-White Hispanic). Of these 45 participants, 42% (*n* = 19) were randomized to the MI condition. Next, a matched sample of self-identified White participants was derived from the White participants who were randomized to MI (*n* = 58). Nineteen of the 58 White participants who were assigned to the MI condition were matched to the 19 POC who were assigned to the MI condition. White participants were matched to POC participants based on POC participants’ age, and income resulting in a sample of 19 POC participants and 19 White participants (*n* = 38). Both samples engaged in similar heavy alcohol use, which precluded the need to include alcohol use as a matching criterion. Participants’ ages ranged from 20 to 60 (*M* = 41.6, SD = 11.4,). Most participants were unemployed (71%), had an annual income less than $20,000 (60.5%), and/or did not have a college education (71.1%). Demographics for the present study are reported in Table 1.

We first compared POC and White participants’ MDS scores: no significant differences emerged between these two groups. Although the average MDS score was not significantly different, POC participants reported greater racial/ethnic discrimination (*M* = 1.90) and HIV status discrimination (*M* = 2.58) compared to White participants (*M* = 1.58; *M* = 1.58; respectively. See Table 1).

### 3.2. MISC Codes

Providers’ global scores when working with POC or White participants were not significantly different (see Table 2). Further, regardless of each participant’s race, providers used similar ratios of reflections to questions and spoke for the same amount of time (see Table 3). Two significant differences emerged, however. We found that, for White versus POC participants, providers asked significantly more open-ended questions (*t* = −2.31, *p* = 0.03) and provided more complex reflections relative to simple reflections (*t* = −2.29, *p* = 0.03; see Table 3). These results suggest that the technical aspects of the brief MI were implemented with less fidelity when the White providers worked with POC participants.

### 3.3. Post-Intervention Drinking

At all follow-ups, both White and POC participants had reduced their alcohol use. There was no significant difference in standard drinks at any follow-up visits between POC and White participants (see Table 4).

Linear mixed effects models were used to examine changes in drinking across the study between BIPOC and White participants (adjusted for baseline weekly standard drinks). Analyses demonstrated that race did not predict the level of change in drinking at follow-ups. There were no significant differences between BIPOC and White participants in the number of drinking days per week (*β* = 13.22; ICC = 0.08; SE = 31.62; CI = 1.31–133.04; *p* = 0.40) nor the number of heavy drinking days at (*β* = 1.88; ICC = 0.05; SE = 2.70; CI = 0.11–31.28; *p* = 0.49) at follow-up visits. These results suggest that the brief MI was equally effective at reducing drinking for both White and POC participants.

### 3.4. Thematic Analyses

Using thematic analysis, the first and second authors qualitatively coded session transcripts, specifically about stigma and discrimination, to better understand session content not captured by the MISC. These analyses revealed that POC and White participants both discussed how stigma tied to their marginalized identities related to their substance use.

**Theme** **1.**POC and White Participants reported drinking alcohol to avoid experiencing negative emotions associated with their sexual identities.

One theme that emerged was participants’ use of alcohol to avoid experiencing negative emotions (e.g., shame) associated with their sexual minority identity. For example: 

Participant:*“I’m comfortable with being gay. Right. I’ve accepted it. I mean, I haven’t any choice. I don’t particularly like it. So, around the social settings, it’s just, sometimes I get depressed. When I go to a gay club, if I’m sober—I’m angry when I go to a gay club because I’m like, ‘Oh, this is gay people, ohh’ And [drinking] makes me feel … I judge, I guess. As soon as I get in there, I start judging… “Oh, a bunch gay people. Oh, they’re probably all promiscuous.” What I’m really… sometimes I’m going in there and I just think negative about it. “What am I doing here?”.”* (Participant: Hispanic Black man)

The previous statement suggests that the participant has internalized stereotypical beliefs about MSM’s sexual behavior, namely that MSM are “promiscuous.” The participant describes using alcohol to facilitate his interactions with other MSM.

A similar example of how alcohol was used to cope with sexual minority identity was expressed by a White participant who described how their family’s values influenced their drinking:

Participant:
*“I don’t come from a—I mean, I come from a middle-class family… My parents are, have basically, done well. My father’s an engineer. And so, but, it’s just crazy. Politics, their politics, their views on the world, and—”*


Provider:
*“So when you came out, things kinda—”*


Participant:*“Yeah, yeah. It’s difficult… It’s difficult. But they just—as I’m, get older—they continue to weigh on me more and more and hold me back, so I, I finally realized—and that’s part of the reason for the alcohol.”* (Participant: non-Hispanic White man)

The previous quote suggests that the participant believes that his socioeconomic background may have contributed to his parents’ lack of support of his queer identity. It could be inferred that the participant’s sexual orientation clashed with their family’s expectations given their race and class. This social pressure may have led the participant to utilize alcohol to cope. This suggests how individuals’ intersecting identities and backgrounds can put unique societal pressures on them.

**Theme** **2.**POC and White participants reported drinking alcohol to avoid experiencing negative emotions associated with their sexual identity HIV status.

Another theme that emerged was how both POC and White participants used alcohol to process their emotions related to their HIV + status. For example, one participant discussed how alcohol helped them feel less angry about their HIV diagnosis:

Provider:
*“Do you feel like your alcohol has increased since you were diagnosed with HIV?”*


Participant:
*“Yeah.”*


Provider:
*“So it sounds like alcohol might be a way for you to help cope with stress or the pain of that.”*


Participant:
*“It’s been a lot better.”*


Provider:
*“What’s been a lot better?”*


Participant:
*“Me. Very angry.”*


Provider:
*“You were very angry.”*


Participant:
*“Yeah.”*


Provider:
*“And you’re less angry now.”*


Participant:
*“I am. I’m less angry.”*
(Participant: Hispanic man)

Another participant articulated how alcohol helped reduce the saliency of their HIV status:

Participant:*“[Alcohol] helps me ignore it there, yeah, so that I can live peacefully with it. Even though I can live peacefully with it, it’s just sometimes I feel like I have the Scarlett Letter. People can look at me and automatically just know that I have it.”* (Participant: Hispanic White man)

During a different session, another participant also articulated how alcohol helped reduce the saliency of their HIV status:

Provider:
*“Then you went through a long period of time where you didn’t drink as much, but then when all this stuff happened, you found out that you had HIV and– it sounds like it was pretty difficult, because you had, for a long while, had been pretending it wa– not getting tested or worried about it. ‘Denial’, I think you used that word.”*


Participant:
*“It’s still– I drink for the same reason, cuz it’s—yeah… it helps with the denial. Sometimes I don’t want to admit I’m sick or live like that. When I drink, it helps just–it started off, ‘Okay. I’ll drink in lieu of doin’ things,’ but then it becomes ‘I’ll drink so I can do things.’”*
(Participant: non-Hispanic White man)

In both quotes above, participants express how alcohol allows them to avoid the perceived stigma of their HIV status. They suggest that both White and POC MSM living with HIV encounter significant social stigma and that the negative affect resulting from perceived stigma about their sexual orientation and HIV status plays a role in their drinking.

**Theme** **3.**POC participants reported experiencing racial discrimination in the context of their substance use.

POC participants described having experienced racial discrimination. POC participants described being treated poorly or exploited for their identities as people of color. For example, one participant expressed:

Participant:
*“Yeah and also, I can’t prove it but I’ve been getting pulled over a lot for no apparent reasons. Whenever I’m driving, I’m always obeying the laws. I put the cruise control on whatever the speed limit is, and they pull me over for that… You can’t prove it, I mean, what can I prove?”*


Provider:
*“Do you have folks in your life that you are able to talk about some of this stuff with?”*


Participant:*“Yeah, I talk to them about it but there’s not much… I’ve been telling everybody who listens because I really feel it’s not fair. I’m not a bad guy. I get railroaded, and I know I’m not the only person. I hate to say it, racism is not dead, because it’s really not.”* (Participant: non-Hispanic Black man)

From another session with a different participant, the provider reflected to the client how racism played a role in the participant’s experiences of substance use: 

Participant:
*“Obviously this is not what I want to do. I do not want to be getting high. Obviously, I know it’s not the right thing to do. Obviously, listen, I go around White people. To be honest with you, I hate Black people. And I know people don’t, that sounds like really awful for me to say—I don’t like Black people. I don’t like to be around crisis people. I don’t like to be around bad situations…”*


Provider:
*“It sounds like there’s so much self-loathing”*


Participant:
*“Absolutely.”*


Provider:
*“… buried under …”*


Participant:
*“Absolutely.”*


Provider:
*“… internalized racism, buried under trauma.”*


Participant:*“Yeah. I’ve been through so much, man. You know what I’m saying?”* (Participant: non-Hispanic Black man)

The previous statement suggests that the participant has internalized racist beliefs about his own identity. In turn, this has made interacting with other Black people difficult. The impact of racism has cut him off from the social support of other people who could potentially have helped him more effectively cope. It is necessary to point out that the brief MI sessions for this study were designed to address individual-level factors associated with alcohol use. Thus, among the quoted participants, discussions of substance use included their experiences of racial discrimination.

## 4. Discussion

In this study, we aimed to determine if racial mismatch between providers and participants impacted the implementation of a brief MI and behavioral outcomes in a sample of HIV-positive heavy drinking MSM. We compared the coded content of brief MI sessions and relevant behavioral outcomes. Our results demonstrated that providers implemented the brief MI differently when working with POC participants. That is, providers asked fewer open-ended questions and used fewer complex reflections when working with POC participants compared to White participants. At face value, these results might imply that providers implemented the brief MI slightly less effectively when working with POC.

Despite the differences in session content, our results suggest that both POC and White participants reduced their alcohol intake following the brief MI. These results are surprising as they suggest that some of the techniques central to MI (i.e., use of complex reflections and open-ended questions) may be less necessary for invoking behavioral change when working with HIV+ MSM of color. Despite providers utilizing these techniques less when working with POC participants relative to White participants, both White and POC participants showed behavioral change. To better understand these results, we utilized qualitative analyses to gain a better understanding of session content not captured by MISC coding.

Results of our qualitative analyses of session transcripts suggest that there was some overlap between discrimination experienced by both POC and White participants. Both groups discussed discrimination associated with their sexual identities and HIV status in the context of their substance use. What was unique to POC participants, however, was how discussions of discrimination based on race emerged as part of the discussion of their drinking. That is, despite the brief MI not being designed to elicit such experiences from participants, nor being directly asked by study providers to discuss such topics, and given that all providers were White (and therefore could be assumed not to share participant’s experiences of racial discrimination), participants organically discussed racism in the context of their problematic drinking. These participant accounts corroborate research suggesting that experiences of sexual orientation-based [44] and race/ethnicity-based [45] discrimination are associated with alcohol use and alcohol-related problems.

Taken together, our results suggest that effectively utilizing relational components of brief MI may be more relevant to engendering change than precisely following technical guidelines when working with HIV+ MSM of color, e.g., by creating a supportive environment where participants are encouraged to explore their own motivations for change. The creation of such a supportive environment likely facilitated discussions of discrimination. Qualitative analyses of sessions demonstrated that the person-centered approach to MI lends itself to exploring issues identified by participants within the context of the intervention. Although not all participants discussed discrimination experiences, our results suggest that, for some HIV+ MSM of color, racial discrimination was linked to their alcohol use. Therefore, the strength of the brief MI may have been its versatility in allowing providers and clients the ability to explore individually meaningful reasons for behavioral change. This versatility may be particularly useful as it can be difficult to adapt behavioral interventions to all possible combinations of marginalized identities. In addition, some research suggests that such adaptation may not be completely necessary for behavioral change [24].

Further, the tenets of MI are consistent with the construct of cultural humility in working with diverse clients. Cultural humility is a multicultural stance towards openness, is other-oriented, involves being self-aware, incorporates self-reflection, and lets the client be the expert in their own experiences (e.g., [46]). These aspects of cultural humility mirror MI’s emphasis on encouraging reflection, patient autonomy, and establishing goals important to the client. Additionally, cultural humility has been shown to facilitate a stronger working alliance and better therapy outcomes with diverse clients [46]. Therefore, it could be that cultural humility’s overlap with MI creates a setting in which clients feel comfortable discussing experiences of discrimination within the context of substance use. This provides further support for an evidence-based behavioral intervention in working with individuals from marginalized groups.

Additionally, resources within clinical settings vary widely. Funding sources may be patchwork and the prioritization of operating costs can make it difficult to implement best practices for clients. Although MI is relatively inexpensive, continued training and supervision of providers may be significant expenses. As such, it may be prudent to prioritize training providers in the most critical components of MI. If limited resources do not allow for ongoing training and supervision of providers, then at least emphasizing the creation of a supportive environment where participants are encouraged to explore their own motivations for change may be a cost-effective way to encourage behavioral change among clients. The present study demonstrates that, even if resources are limited and it is not feasible to culturally tailor MI interventions or hire POC providers, adhering to the bedrock principles of MI can yield results and create change.

To this end, the authors offer several suggestions for providers and those clinical supervisors if resources for training/supervision are limited. First, we suggest prioritizing patient autonomy. Specifically, clinicians can emphasize to the client that it is their own personal choice whether to change their behavior and that they can choose how such change occurs. In addition, we suggest prioritizing displaying empathy with clients. By actively listening, providers can come to understand the client’s perspective (e.g., situation, and emotions). The provider can utilize reflections and exploratory questions as needed to gain a deeper understanding of the client’s perspective. Finally, we advocate for providers to be receptive to patients’ experiences. That is, especially for White providers, to listen to clients’ accounts of racial/ethnic discrimination and allow them to explore the impact such experiences have on their daily lives. We also recommend perusing relevant literature on how mental health care disparities impact POC sexual and gender minority individuals [47].

### Limitations

We acknowledge that MI cannot address the fundamental causes of heath disparities among POC populations. Indeed, no individual-level intervention could do so given the embedded, structural nature of the problems. Systems of oppression and violence operating within the U.S. play a direct role in worse health among POC groups. In addition, discrimination within healthcare settings contributes to POC’s reduced access to care [47]. MI may be useful in modifying specific behaviors within individual members of disadvantaged populations. Further, we advocate for the dismantling of racist and heteronormative institutions that perpetuate these disparities.

The current study was a secondary data analysis of a larger clinical trial, which limited the scope of variables and analyses. The number of participants and composition of the sample need to be considered as these may impact our findings. For example, although we did not find significant differences in changes to drinking behavior (e.g., reductions in drinks per week) between POC and White participants, such differences were rather large. POC participants reduced the number of drinks they consumed from 30.9 standard drinks at baseline to 8.8 at 12 months, whereas White participants dropped from 22 to 10.6 at 12 months. Such results suggest that the small sample and resultant increase in variability between POC and White participants may have impacted our ability to detect significant differences in drinking at follow-up visits.

The lack of diversity in our sample also limits the generalizability of our findings. For example, the sample was predominantly White, restricting the size of the racial minority group we could analyze. Additionally, the majority of POC participants identified as Black, further limiting inferences we can draw concerning the larger community of MSM of color. Further, we did not have diversity regarding socioeconomic status, as most of the participants were unemployed, most did not have a college degree or higher, and the majority had an annual income less than $20,000. More research is needed with larger and more diverse samples regarding the utility of MI with different populations.

The lack of diversity in socioeconomic status of the sample also impacted the matching procedure we utilized. Our goal was to compare all 19 POC participants who received the MI intervention with a socio-demographically matched sample of White participants. Despite utilizing matching procedures, discrepancies arose (e.g., 15% of POC participants had attained at least a college degree compared to 47% of White participants.) All such differences were not significant, but still suggest that the groups were not completely comparable. Given that the current research was a secondary data analysis of a larger clinical trial, we could not recruit a more diverse sample to address limitations in participant recruitment. Future research would do well to further examine how the intersection of socio-economic status and racial/ethnic identities may influence drinking behavior.

In addition, the fact that we did not find significant differences between White and POC participants on the racial/ethnic discrimination subscale of the MDS points to potential measurement limitations. One would expect self-identified White participants to endorse less racial discrimination than POC participants. Nevertheless, some White participants in our sample reported experiencing racial/ethnic discrimination. These findings may be due to the MDS’ conflation of race and ethnicity. For example, White participants might have completed the racial discrimination subscale of the MDS while considering possible discrimination within the context of belonging to a religious minority (e.g., Judaism) or membership in an ethnic minority group (e.g., Middle Eastern or North African). Given this ambiguity, we decided to retain these participants’ data. Future studies should intentionally assess and differentiate between race and ethnicity, to examine multiple forms of marginalization associated with participants’ identities.

Despite these limitations, our findings highlight the utility of MI across racial/ethnic groups and underscore the importance of targeting discrimination within the context an alcohol related motivational intervention.

## 5. Conclusions

The current study suggests that motivational interviewing can provide a therapeutic space for HIV-positive cisgender POC MSM to discuss experiences of discrimination even when it is not the primary aim of the intervention. Our findings suggest that clinicians should be prepared to discuss broader systems of oppression for patients with marginalized identities as these experiences may be intertwined with other psychosocial stressors.

## Figures and Tables

**Table 1 ijerph-19-03930-t001:** Baseline characteristics of the sample (demographics and key variables).

Variable (*n* = 38)	Total (*n* = 38) Mean (SD) or N (%)	Persons of Color (POC; *n* = 19) Mean (SD) or N (%)	White (*n* = 19) Mean (SD) or N (%)	*t* or *χ*^2^	*p*
Age (Range: 20–60)	41.6 (11.4)	40.26 (11.6)	42.8 (11.4)	0.69	0.49
Ethnicity					
Hispanic or Latino	10 (26.3)	6 (31.6)	4 (21.1)	0.54	0.71
Race					
American Indian/Alaskan Native	1 (2.6)	1 (2.6)	0 (0.0)	1.03	0.31
Black or African American	15 (39.5)	15 (39.5)	0 (0.0)	24.78	>0.01
White	19 (50.00)	0 (0.0)	19 (100.0)	38.00	>0.01
College degree or more	11 (28.9)	3 (15.8)	8 (47.2)	3.2	0.07
Annual Income < $20,000	23 (60.5)	13 (68.4)	10 (52.6)	0.99	0.32
Unemployed	26 (68.4)	14 (73.7)	12 (63.2)	0.49	0.49
Identify as gay or bisexual	36 (94.7)	18 (94.7)	18 (94.7)	0.00	1.00
Multiple Discrimination Scale (MDS) scores ^1^					
Race/Ethnicity Discrimination	1.74 (2.89)	1.89 (2.73)	1.58 (3.11)	−0.33	0.74
Sexual Orientation Discrimination	2.08 (2.66)	2.58 (2.57)	1.58 (2.73)	−1.16	0.25
HIV Status Discrimination	1.53 (1.97)	1.53 (2.01)	1.53 (1.98)	−0.00	1.00

^1^ Each subscale of the MDS had a possible score range from 0 (no events endorsed) to 10 (all events endorsed).

**Table 2 ijerph-19-03930-t002:** MISC Global scores.

Variable (*n* = 38) ^1^	Total (*n* = 38) Mean (SD)	Persons of Color (POC; *n* = 19) Mean (SD)	White (*n* = 19) Mean (SD)	Cohen’s d	T	*p*
Provider Acceptance	3.9 (0.8)	3.7 (0.9)	4.2 (0.6)	0.63	−1.93	0.06
Provider Empathy	4.0 (0.7)	3.8 (0.8)	4.1 (0.7)	0.44	−1.34	0.18
Provider Directiveness	3.8 (0.8)	3.7 (0.9)	4.0 (0.7)	0.34	−1.01	0.32
Provider Respect for Autonomy	3.8 (0.7)	3.7 (0.9)	3.8 (0.4)	0.23	−0.72	0.48
Provider Overall Collaborativeness	3.6 (0.8)	3.6 (0.9)	3.7 (0.7)	0.13	−0.43	0.67
Provider Evocation	3.9 (0.6)	3.7 (0.5)	4.0 (0.7)	0.46	−1.42	0.16

^1^ All variables had a possible range from 1–5.

**Table 3 ijerph-19-03930-t003:** MISC Utterances proportions.

Variable (*n* = 38)	Total (*n* = 38) Mean (SD)	Persons of Color (POC; *n* = 19) Mean (SD)	White (*n* = 19) Mean (SD)	Cohen’s d	T	*p*
Provider Proportion open Questions	0.34 (0.09)	0.31 (0.08)	0.37 (0.09)	0.67	−2.31	0.03
Provider Proportion Complex Reflections	0.71 (0.11)	0.67 (0.11)	0.75 (0.10)	0.76	−2.29	0.03
Provider Proportion Directives	0.15 (0.04)	0.15 (0.04)	0.16 0(.03)	0.28	−1.00	0.33
Ratio of Providers Reflections to Questions	1.15 (0.04)	1.04 (0.48)	1.27 (0.62)	0.41	−1.30	0.20
Number of Provider Utterances	93.0 (22.47)	88.68 (23.66)	97.21 (21.47)	0.38	−1.16	0.25
Provider Percentage of all Utterances	13.99 (3.16)	13.63 (3.04)	14.34 (3.31)	0.22	−0.69	0.50

**Table 4 ijerph-19-03930-t004:** Drinking behavior across study visits.

Variable (*n* = 38)	Total (*n* = 38) Mean (SD)	Persons of Color (POC; *n* = 19) Mean (SD)	White (*n* = 19) Mean (SD)	*t*	*p*
Drinks per week					
Baseline	26.5 (27.4)	30.9 (35.6)	22.2 (15.2)	−0.98	0.33
3 months	18.1 (22.2)	17.9 (29.1)	18.2 (12.9)	0.03	0.26
6 months	10.3 (11.1)	8.5 (11.4)	12.2 (10.8)	1.00	0.33
12 months	9.6 (12.6)	8.8 (2.9)	10.6 (9.5)	0.42	0.68
Heavy drinking days per month					
Baseline	8.8 (7.6)	9.1 (8.1)	8.6 (7.2)	−0.21	0.83
3 months	6.4 (7.9)	5.6 (8.3)	7.2 (7.6)	0.58	0.56
6 months	3.9 (6.2)	2.5 (4.3)	5.4 (7.6)	0.1.4	0.17
12 months	2.6 (4.8)	1.8 (3.3)	3.4 (6.1)	0.93	0.36

## Data Availability

The data presented in this study are available upon request.

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
