# Peer review of "Examining the Impact of Race on Motivational Interviewing Implementation and Outcomes with HIV+ Heavy Drinking Men Who Have Sex with Men"

_ijerph, 2022, doi:10.3390/ijerph19073930_

Round 1

Reviewer 1 Report

In this study, Surace et al. explored the role of race on the implementation and the impact of motivational interviewing sessions among MSM living with HIV who reported heavy drinking. Authors found that MI providers asked more open-ended questions and more complex reflections when interviewing white individuals. However, no differences in alcohol drinking reduction were observed between the 2 groups. 

This work addresses an important aspect of care for people living with HIV. The article is very well structured and nicely written. The mixed-method approach is very strong, and the themes identified in the qualitative analysis reveal important insights for the care of people living with HIV.  I have a few minor comments to further strengthen the manuscript: 

  1. Language: Please refer to the 2015 UNAIDS Terminology Guidelines for preferred wording (https://www.unaids.org/en/resources/documents/2015/2015_terminology_guidelines). For instance, “men who have sex with men” is preferred over gay, and “people (living) with HIV” is preferred over HIV positive or HIV infected. In addition, the term “Latinx” has led to a controversy, as only 3% of Latin-American individuals identify themselves as Latinx (see https://osf.io/m39v5/ for a nice discussion of the subject). 
  2. Table 1: The horizontal lines are not correct. For instance, in its current form, College degree, income, and employment status are summarized within Race. 
  3. Please report actual p-values, not only <0.05.  
  4. Tables: Please provide explanations for abbreviations.

Author Response

1. Language: Please refer to the 2015 UNAIDS Terminology Guidelines for preferred wording (https://www.unaids.org/en/resources/documents/2015/2015_terminology_guidelines). For instance, “men who have sex with men” is preferred over gay, and “people (living) with HIV” is preferred over HIV positive or HIV infected. In addition, the term “Latinx” has led to a controversy, as only 3% of Latin-American individuals identify themselves as Latinx (see https://osf.io/m39v5/ for a nice discussion of the subject). 

  • We thank the reviewer for their comments and agree that the language can be changed to better reflect the current conventions. We have removed the terms “gay” and “Latinx” from the non-quoted content of the manuscript.

2. Table 1: The horizontal lines are not correct. For instance, in its current form, College degree, income, and employment status are summarized within Race. 

  • We thank the reviewer for pointing this out to us. We have reformatted the table to correct this issue.

3. Please report actual p-values, not only <0.05.  

    • We have edited the results and now report exact p values.

4. Tables: Please provide explanations for abbreviations.

We have clarified the meaning of abbreviations (e.g., POC). 

Reviewer 2 Report

The authors used secondary data and applied a mixed method approach to assess how racial mismatch between clients and providers may impact MI implementation and subsequent behaviors. From implementation science point of view, it is helpful to identify strategies to enhance the effectiveness of the (MI) intervention.

There are two main hypotheses (line 110, 114) in this manuscript: 1. For white (racial matched), provider would adhere more closely to technical and relational components of MI compared to POC (racial mismatching). 2. Sessions between white (racial matching) would result in more behavior change in post-intervention alcohol consumption. I suggest the authors to better specify the outcome measures and statistical approaches taken.

  • Line 139-141, specify the possible ranges of these 3 composite measures.
  • Line 199-208, quantitative analysis section, consider including the statistical approaches used and statistics estimated reported.
  • For comparison of adherence to MI principle and providers’ session utterances,
  • Line 189, please better specify these measures presented in Table 2 and Table 3 including the possible ranges to help readers to understand these measurements. Are these composite measures? Does the higher value indicate better outcome?
  • Similarly, for Table 2 and Table 3, if I am not mistaken, these are continuous outcomes. Please remove N (%) and χ2 to avoid confusion.
  • Were multiple comparison adjustments considered?
  • For the comparison of drinking behaviors,
    • Line 208-210, longitudinal analysis (i.e., mixed model approach) is normally used to examine changes at multiple time points. The author should consider using longitudinal data analysis to analyze participants’ changes in drinking behavior at different study follow up period (baseline, 3, 6 and 9 months). Alternatively, changes from the baseline (at 3, 6 and 12) can be considered (to better align with the hypothesis).
    • Line 253-257 (table 1), the results seem to suggest that there is a larger reduction in drinks per week or heavy drinking days per month in the POC group compared with White. Based on Table 1, the reduction is 21 drinks per week among POC vs 11 drinks among White at 12 months; the reduction is 7.3 heavy drinking days among POC vs 5.2 heavy drinking days per month among White.
    • Please revise the results and discussion accordingly based on the updated analysis results.
    • Consider separating the results of the main outcome measures (drinking behaviors) from the baseline characteristics and to avoid confusing.
  • Line 240 (Table 1),
    • Better label the table (i.e., separating mean (SD) N (%); T or χ2)
    • On line 227, the author stated that “White participants were matched to POC participants’ age, income and education.” Yet, there is still a big difference (15% vs 47%) between POC and white in participants with college degree or more. Please check.

Author Response

  1. Line 139-141, specify the possible ranges of these 3 composite measures.
    • We have clarified that each subscale had a possible score range from 0 (no events endorsed) to 10 (all events endorsed).

  1. Line 199-208, quantitative analysis section, consider including the statistical approaches used and statistics estimated reported.

    • We thank the reviewer for their comment and agree that including statistical approaches used and estimates would improve the quality of the manuscript and have edited the quantitative analysis section accordingly.

  1. For comparison of adherence to MI principle and providers’ session utterances,

Line 189, please better specify these measures presented in Table 2 and Table 3 including the possible ranges to help readers to understand these measurements. Are these composite measures? Does the higher value indicate better outcome?

    • We agree the tables could be edited for clarity. We have added subscripts to the tables
      to improve their interpretation (e.g., including score ranges).

  1. Similarly, for Table 2 and Table 3, if I am not mistaken, these are continuous outcomes. Please remove N (%) and χ2 to avoid confusion.

    • We have removed these notations from the table.

  1. Were multiple comparison adjustments considered?

    • The authors had not considered using multiple comparison adjustments. Our rationale for not making such adjustments was that we were not certain which approach best suited our data. In addition, given that only two quantitative hypotheses were being tested we did not feel the risk of committing a type one error was high.  

  1. For the comparison of drinking behaviors,

Line 208-210, longitudinal analysis (i.e., mixed model approach) is normally used to examine changes at multiple time points. The author should consider using longitudinal data analysis to analyze participants’ changes in drinking behavior at different study follow up period (baseline, 3, 6 and 9 months). Alternatively, changes from the baseline (at 3, 6 and 12) can be considered (to better align with the hypothesis).

      • The authors thank the reviewer for their comment- we had not previously considered conducting these analyses and agree that including such analyses would strengthen the manuscript. We conducted multilevel modeling and included the following results in the revised manuscript:

Multilevel modeling was used to examine the changes in drinking across the study between BIPOC and White participants. Analyses demonstrated that race did not predict the level of change in drinking at follow-ups. There were no significant differences between BIPOC and White participants in the number of drinking days per week (β=13.22 ICC=.08 SE=31.62 CI=1.31 - 133.04 p= .49) nor the number of heavy drinking days at (β =1.88 ICC=.05 SE=2.70 CI=.11 - 31.28 p= .49) at follow-up visits.  

Line 253-257 (table 1), the results seem to suggest that there is a larger reduction in drinks per week or heavy drinking days per month in the POC group compared with White. Based on Table 1, the reduction is 21 drinks per week among POC vs 11 drinks among White at 12 months; the reduction is 7.3 heavy drinking days among POC vs 5.2 heavy drinking days per month among White.

      • The authors acknowledge that a difference in drinks per week was found between POC and White participants. Our multilevel model suggests that race was not predictive of significant differences in drinking behavior at follow-up (see our revised results and discussion sections).

Please revise the results and discussion accordingly based on the updated analysis results.

      • The authors have revised the results and discussion sections to reflect the updated analyses.

Consider separating the results of the main outcome measures (drinking behaviors) from the baseline characteristics and to avoid confusing.

      • The authors agree that separating the demographic and drinking behavior results would improve the manuscript. Therefore, we have created a separate table specifically for drinking behavior across study visits (see Table 4 of the revised manuscript).

  1. Line 240 (Table 1),

Better label the table (i.e., separating mean (SD) N (%); T or χ2)

      • The authors have revised the tables for clarity.
  1. On line 227, the author stated that “White participants were matched to POC participants’ age, income and education.” Yet, there is still a big difference (15% vs 47%) between POC and white in participants with college degree or more. Please check.

    • The authors thank the reviewer for pointing this out to us. We agree that there is a discrepancy in education between White and POC participants. We verified the matching procedures and can confirm that the sampled White participants were the closest we could come to matching the socio-economic status of our POC participants. We have expanded the limitations section to include a discussion of discrepancies in matching due to sample size constraints.

Reviewer 3 Report

The article is focused on the question of race-influenced differences in Motivational Interviewing implementation and subsequent effects in client engagement in intervention and behavioural change. The authors obtained interesting facts of how drinking behaviour intertwines with clients’ experience of discrimination based on HIV+ status and non-normative (MSM) sexual practice. The use of both qualitative and quantitative analysis within the research deserves a favourable estimation: due to this combination, the research produces some new knowledge and facilitates proper understandings of the wide spectrum of meaningful reasons for behavioural change when counselling minoritized individuals. The article is valuable precisely for its empirical data. First, it is notable that although racial mismatch between counsellor and client does impact the implementation of a brief Motivational Interviewing, and this impact relates to essential technical proficiencies such as asking open-ended questions and providing complex reflections, the intervention outcomes turn to be equally effective. This fact supports the general idea that the very relational aspect is crucial for obtaining positive outcomes in any kind of psychological intervention. And the thematic analysis demonstrates the leading role of speaking out in supportive setting for invoking behavioural change. Second, the qualitative data of post-intervention sessions describe the grounds of excessive drinking that are to be in focus of help providers for minoritized individuals. Sharing of such knowledge is an important part of improving helping competencies.

In my opinion, what is missing in the introductory part is the theoretical, not empirical, rationale for suggestion that the very racial identity mismatching between the help provider and the client may evoke less fidelity in implementing the technical procedures of brief MI. Providers may be unconscious of different implementation of MI when dealing with clients with different identities – either racial or sexual and so on. But there ought to be some socio-psychological model explaining why providers, racially matched for their clients, ask more open-ended questions and deliver a higher complex reflection. Otherwise, this hypothesis looks like retroactively advanced. And this is the only comment to the submitted manuscript.

Author Response

  1. In my opinion, what is missing in the introductory part is the theoretical, not empirical, rationale for suggestion that the very racial identity mismatching between the help provider and the client may evoke less fidelity in implementing the technical procedures of brief MI. Providers may be unconscious of different implementation of MI when dealing with clients with different identities – either racial or sexual and so on. But there ought to be some socio-psychological model explaining why providers, racially matched for their clients, ask more open-ended questions and deliver a higher complex reflection. Otherwise, this hypothesis looks like retroactively advanced. And this is the only comment to the submitted manuscript.

  • The authors thank the reviewer for their comment and agree more of a rationale may be warranted for our hypotheses. We have edited the Introduction to include a discussion of racial bias and how it may influence he provider/client relationship.

Round 2

Reviewer 2 Report

The authors have enhanced the manuscript. There are remaining concerns about the analysis. Please see below.

Line 485, What are the multilevel models? For longitudinal data, the observations from the same subject at different time points are correlated. The linear mixed model approach accounting for the correlations within a subject is often used for the continuous outcomes. Is the multilevel model here referred to the mixed model approach here? Were the baseline values adjusted?  Please clarify.

Line 502, How many times were the sample matched?  The total sample size is 180. If there are 45 self-identified as a POC, the remaining 135 are White. Line 503, The authors stated, “a matched sample of self-identified white….”, what were matched here? Were there only 58 out of 135 matched and were randomized to MI? Line 504, the authors stated, “White participants were matched to POC on age, income, and education, resulting in a sample of 19…”.  Is this the second time matching? Were 19 out of 58 White matched to POC? 

Line 505, Based on the authors' response, education was not matched.  Therefore, education should be removed here.

Line 533, Table 1, about the race, their number for POC does not sum up to 19 and the percentage does not sum up to 1. Is there a missing category in RACE? The Chi-squared test for RACE should yield only one p-value. 

Line 533, Table 1, for HIV status discrimination, the mean (sd) for POC is 2.58 (2.57) and for White is 1.58 (2.7), yet the p-value is equal to 1. Please check.

Line 717-718, Please interpret β. Is β the mean difference overtime? These two p-values should not be the same. The p-value (line 717) does not seem to be correct. Please check. How do the results here link to Table 4?

Line 721, Table 4, The N(%) should be removed from the label. Based on the table, the reduction of drinks per week for POC was 22, dropped from 30.9 at baseline to 8.5 at the 6 months, for white was 10, dropped from 22 to 12 at 6 months. The reduction was 12 drinks (2 times) higher in POC compared to White. Similar results were observed at 12 months. What are the possible reasons?  Similar patterns were also found for the outcome of heavy drinking days per month. Although the p-value is not significant, it is likely due to the small sample size.

The authors should consider removing t or χ2 values in the manuscript since these values normally are not reported.

Author Response

  1. Line 485, What are the multilevel models? For longitudinal data, the observations from the same subject at different time points are correlated. The linear mixed model approach accounting for the correlations within a subject is often used for the continuous outcomes. Is the multilevel model here referred to the mixed model approach here? Were the baseline values adjusted?  Please clarify.

  • We have edited the Results section to clarify the type of multilevel modeling used and adjustments made. We wrote that “Linear mixed effects models were used to examine changes in drinking across the study between BIPOC and White participants (adjusted for baseline weekly standard drinks).”

  1. Line 502, How many times were the sample matched?  The total sample size is 180. If there are 45 self-identified as a POC, the remaining 135 are White.

  • The sample was matched once.

  1. Line 503, The authors stated, “a matched sample of self-identified white….”, what were matched here? Were there only 58 out of 135 matched and were randomized to MI?

  • Yes, 58 of the 135 White participants were randomized to MI. To make this more explicit line 235 now reads “Next, a matched sample of self-identified White participants was derived from the White participants who were randomized to MI (n=58).”

  1. Line 504, the authors stated, “White participants were matched to POC on age, income, and education, resulting in a sample of 19…”.  Is this the second time matching? Were 19 out of 58 White matched to POC? 

  • No, the sample was matched once using SPSS matching syntax. Yes, 19 of the 58 White participants who were assigned to MI were matched to the 19 POC who were assigned MI. We have edited the Results to now read “Nineteen of the 58 White participants who were assigned to the MI condition were matched to the 19 POC who were assigned to the MI condition. White participants were matched to POC participants based on POC participants’ age, income, and education, resulting in a sample of 19 POC participants and 19 White participants (n=38).”

  1. Line 505, Based on the authors' response, education was not matched.  Therefore, education should be removed here.

  • The authors recognize that our matching procedures were not ideal. We have removed education as one of our matching criteria from the manuscript.

  1. Line 533, Table 1, about the race, their number for POC does not sum up to 19 and the percentage does not sum up to 1. Is there a missing category in RACE? The Chi-squared test for RACE should yield only one p-value. 

  • The reason the Race categories do not sum up to 19 is because 3 of the POC participants identified as Hispanic or Latino and not White. We define POC in the Results section as self-identified “Black/African American, American Indian/Alaskan Native, Asian and/or non-White Hispanic”. The reason we included more than 1 chi-squared test was because 4 of our White participants also identified as Hispanic/Latino.    

  1. Line 533, Table 1, for HIV status discrimination, the mean (sd) for POC is 2.58 (2.57) and for White is 1.58 (2.7), yet the p-value is equal to 1. Please check.

  • The authors thank the reviewer for catching this error. The first author mistaken entered the wrong values for MDS scores in Table 1. The revised manuscript now has the correct values.

  1. Line 717-718, Please interpret β. Is β the mean difference overtime?

  • The beta values correspond to predicted differences in standard drinks at follow-up. Specifically in these models Race was dummy coded (0=White participants and 1=POC participants), therefore these betas correspond to the predicted change in standard drinks consumed over time of POC participants (White was the referent).

  1. These two p-values should not be the same. The p-value (line 717) does not seem to be correct. Please check.

  • The authors thank the reviewer for pointing this out. The first author entered the wrong values for the p-values. The correct values have been entered.

  1. How do the results here link to Table 4?
    • The longitudinal analysis (i.e., mixed effects models) are meant to examine changes in drinking behavior at multiple time points. These modeling results are meant to further demonstrate the non-significant differences in drinking behavior at different study follow up period shown in table 4. We have separated the results of the mixed effects models from the description of Table 4 to avoid confusion.

  1. Line 721, Table 4, The N(%) should be removed from the label.

  • The authors have removed “N(%)” from Table 4.

  1. Based on the table, the reduction of drinks per week for POC was 22, dropped from 30.9 at baseline to 8.5 at the 6 months, for white was 10, dropped from 22 to 12 at 6 months. The reduction was 12 drinks (2 times) higher in POC compared to White. Similar results were observed at 12 months. What are the possible reasons?  Similar patterns were also found for the outcome of heavy drinking days per month. Although the p-value is not significant, it is likely due to the small sample size.

  • The authors acknowledge that differences in drinking outcomes across time between White and POC participants were non-significant, nevertheless such differences were quite pronounced. The authors agree such differences were likely due to sample size limitations. We have expanded our limitations section to address the issues related to sample size. Paragraph two of the Limitations section now reads:

The current study was a secondary data analysis of a larger clinical trial, which limited the scope of variables and analyses. The number of participants and composition of the sample need to be considered as it may impact our findings. For example, although we did not find significant differences in changes to drinking behavior (e.g., reductions in drinks per week) between POC and White participants, such differences were rather large. POC participants reduced the number of drinks they consumed from 30.9 standard drinks at baseline to 8.8 at the 12 months, whereas White participants dropped from 22 to 10.6 at 12 months. Such results suggest that the small sample and resultant increase in variability between POC and White participants may have impacted our ability to detect significant differences in drinking at follow-up visits.

  1. The authors should consider removing t or χ2 values in the manuscript since these values normally are not reported.

  • The authors have elected to keep t or χ2 values to increase clarity or findings and ensure understanding.